# Role of PRMT1 and PRMT5 in Breast Cancer

**DOI:** 10.3390/ijms25168854

**Published:** 2024-08-14

**Authors:** Sébastien Martinez, Stéphanie Sentis, Coralie Poulard, Olivier Trédan, Muriel Le Romancer

**Affiliations:** 1Inserm U1052, Centre de Recherche en Cancérologie de Lyon, Université Claude Bernard Lyon 1, F-69000 Lyon, France; 2CNRS UMR5286, Centre de Recherche en Cancérologie de Lyon, Université Claude Bernard Lyon 1, F-69000 Lyon, France; 3Oncology Department, Centre Leon Bérard, F-69008 Lyon, France

**Keywords:** PRMT1, PRMT5, arginine methylation, breast cancer, transcriptional regulation, cell signaling

## Abstract

Breast cancer is the most common cancer diagnosed in women worldwide. Early-stage breast cancer is curable in ~70–80% of patients, while advanced metastatic breast cancer is considered incurable with current therapies. Breast cancer is a highly heterogeneous disease categorized into three main subtypes based on key markers orientating specific treatment strategies for each subtype. The complexity of breast carcinogenesis is often associated with epigenetic modification regulating different signaling pathways, involved in breast tumor initiation and progression, particularly by the methylation of arginine residues. Protein arginine methyltransferases (PRMT1-9) have emerged, through their ability to methylate histones and non-histone substrates, as essential regulators of cancers. Here, we present an updated overview of the mechanisms by which PRMT1 and PRMT5, two major members of the PRMT family, control important signaling pathways impacting breast tumorigenesis, highlighting them as putative therapeutic targets.

## 1. Introduction

Breast cancer (BC) is the most common cancer in women worldwide, affecting 1 in 8 women over the course of their lives [1,2]. BC is a heterogeneous disease classified into different subtypes according to histological/molecular biomarkers used to determine treatment strategies. Luminal BC is characterized by the expression of estrogen receptor (ERα) and/or progesterone receptor (PR); HER2^+^ BC is defined by the overexpression of the human epidermal growth factor (HER2), and triple-negative BC (TNBC) encompasses tumors lacking ERα/PR and HER2 expression [3,4,5]. Patients with (i) luminal BC are treated with endocrine therapy such as tamoxifen for premenopausal patients and aromatase inhibitors for postmenopausal patients [6]. Fulvestrant, another anti-estrogen, is used in the metastatic setting [7]. (ii) HER2 BCs are treated with targeted therapies such as monoclonal antibodies or antibody drug conjugates [5], and (iii) TNBCs are treated with conventional chemotherapy [8]. Despite improvement in patient survival over the last decades, resistance occurs in 30% of cases for multiple causes that are still a matter of research [9]. Hence the need to define biomarkers predictive of response to treatment and to identify new targets is an intensive research topic.

Arginine methylation has emerged as a major post-translational modification (PTM) of histone and non-histone proteins, regulating their expression and biological properties [10]. The protein arginine methyltransferase (PRMT) family comprises nine members classified according to the type of methylation they catalyze. PRMTs are frequently overexpressed in cancer, and their upregulation is often associated with poor prognosis [11,12]. This is particularly true for PRMT1 and PRMT5, the two most extensively studied PRMTs, which play an important role in breast tumorigenesis and resistance, making them promising drug targets. The aim of this review is to summarize current and emerging knowledge on the roles of PRMT1 and PRMT5 in BC and how targeting their methyltransferase activities specifically in BC subtypes could provide new therapeutic tools. 

## 2. Protein Arginine Methyltransferases

Arginine methylation is the addition of methyl groups from S-adenosylmethionine (SAM) to the guanidino nitrogen of arginine side chains. It was first documented in the late 1960s and early 1970s [13,14]. This enzymatic reaction is regulated by members of PRMT family. To date, nine PRMTs (PRMT1–9) have been identified in mammalian cells since the discovery of the first member, PRMT1, in 1996 [10,15]. There are three types of methylated arginine residues in mammalian cells: ω-NG-monomethylarginine (MMA); ω-NG, NG-asymmetric dimethylarginine (ADMA); and ω-NG, N’G-symmetric dimethylarginine (SDMA). PRMTs are classified into three types based on the methylation they catalyze. Type I PRMTs (PRMT1, PRMT2, PRMT3, Carm1 (PRMT4), PRMT6, and PRMT8) catalyze both MMA and ADMA [16]. Type II PRMTs (PRMT5 and PRMT9) generate MMA and SDMA [17]. PRMT7, the only Type III enzyme, exclusively catalyzes the formation of MMA [18]. These modifications play crucial roles in various biological processes, demonstrating the dynamic nature of arginine methylation, which can be reversed by arginine demethylases.

The diversification of proteomes heavily relies on PTMs to determine protein conformation, subcellular localization, interaction with other proteins, stability, and/or activity. As such, arginine methylation participates in various essential cellular processes, maintaining tissue homeostasis [17,19]. PRMTs can dynamically regulate chromatin structure and work as coregulators to activate or repress transcription and thus gene expression. PRMTs can deposit histone marks that either activate (H4R3me2a, H3R2me2s, H3R17me2a, and H3R26me2a) or repress (H3R2me2a, H3R8me2a, H3R8me2s, and H4R3me2s) transcription [20]. As such, PRMTs have a significant impact on various cellular processes, including growth, proliferation, differentiation, transcription, DNA damage and repair, the immune system, RNA processing/alternative splicing, and signal transduction. Accumulating evidence suggests that arginine methylation may be a key factor in the development, progression, and aggressiveness of various types of cancers [12,21]. Although the connection between arginine methylation and cancer is a relatively new research area, studies conducted so far on PRMTs, and cancers have demonstrated their tight link. Indeed, dysregulated PRMT expression has been reported in various human tumors, including lung, breast, prostate, colorectal, bladder tumors, and leukemia [12,22]. For instance, PRMT1, which is responsible for over 90% of arginine methylation, has been shown to promote migratory and invasive behaviors in BC cells by mediating the asymmetric dimethylation of R3 residue of histone H4 (H4R3me2as) at the ZEB1 promoter, thus inducing processes linked to epithelial–mesenchymal transition (EMT) [23]. PRMT5, the major type II enzyme, targets genes involved in BC cell proliferation and invasion and is associated with poor clinical outcome [24,25]. However, its targeting is more complex as PRMT5 activity differs according to its subcellular localization in the different subtypes. Given that PRMT1 and PRMT5 are currently the PRMTs known to contribute the most to BC development and treatment resistance, the primary focus of this review is to specifically highlight the current knowledge regarding their involvement in BC and the emerging therapeutic potential of targeting these enzymes to control disease recurrence, metastasis, and clinical outcome according to BC subtype.

## 3. PRMT1 in Breast Cancer

PRMT1 is a ubiquitous protein in the body, responsible for the majority of ADMA on arginine residues. PRMT1 activity is essential during development, notably for cardiovascular, pancreatic, and brain development, and its knockout is lethal in mice [26,27,28]. In many cancers types, PRMT1 is overexpressed, and its overexpression is often correlated with cancer grade and poor prognosis [11,12,21,29,30,31]. This is also true for BC, particularly for luminal tumors where a high expression of PRMT1 is associated with a decrease of relapse-free survival (Figure 1). Inhibiting PRMT1 activity has been shown to hinder the growth of solid tumors such as lung tumors, ovarian cancers, and hematological and breast tumors [32]. 

Due to the diversity of its histone and non-histone substrates such as tumor suppressors, oncogenes, and steroid receptors [23,34,35,36,37,38] (Table 1), it is not surprising that deregulation of PRMT1 arginine methyltransferase activity may contribute to BC initiation and progression [39,40]. 

### 3.1. Regulation of Tumor Suppressor Pathways

By methylating tumor suppressor genes, PRMT1 inhibits their inhibitory function and promotes the proliferation and growth of BC cells. PRMT1 directly interacts with p53, and PRMT1-mediated p53 methylation inhibits its transcriptional activity, preventing BC cells from undergoing apoptosis or entering senescence, and accelerates proliferation. Arginine-methylated p53 is found in BC patients, and the methylation signal of p53 can be weakened by silencing PRMT1 with shRNA or inhibiting PRMT1 activity with specific inhibitors [44]. In BC, several arginine mutations have been identified in the DNA-binding domain of p53. These mutations decrease p53 DNA binding and transcriptional abilities [41,42]. 

BRCA1 is a nuclear–cytoplasmic protein, and its function differs in the different cellular compartments [43]. In the cytoplasm, BRCA1 promotes BCL2 proteasome-dependent degradation, while nuclear BRCA1 activates the homologous recombination (HR) pathway [45,46] and also regulates the transcription of genes such as BCL2 [34]. PRMT1 methylates BRCA1 on R610 residue in BC cell lines and in breast tumors, influencing its transcriptional cofactor functions [34]. In response to irradiation, SAM levels are increased, inducing methylation of BRCA1 by PRMT1, leading to its nuclear translocation and therefore avoiding cytosolic BRCA1/BCL2 association. This results in BCL2 stabilization and its accumulation in mitochondria, leading to inhibition of apoptosis [47]. By modulating the cellular localization of BRCA1, PRMT1 is an important regulator of the oncogenic functions of BRCA1, contributing to the epigenetic defense of BC cells against ionizing radiation [47] (Figure 2). 

CCAAT/enhancer-binding protein α (C/EBPα) is an essential transcription factor with anti-proliferative activity in BC. In concert with the histone deacetylase HDAC3, C/EBPα binds to the promoter of cyclin D1 and inhibits its expression [48,49]. Methylation of C/EBPα at R35, R156, and R165 residues by PRMT1 prevents its interaction with HDAC3, promotes the expression of cyclin D1, and increases the proliferation of BC cells. [40] (Figure 2). 

### 3.2. Regulations of Oncogenic Pathways

The Wnt signaling pathway regulates key tumor-associated characteristics such as migration, proliferation, and chemoresistance. In TNBC, the Wnt signaling pathway is hyper-activated and promotes tumor progression through the overexpression of Frizzled and its co-receptors, LDL Receptor-Related Protein 5 (LRP5) and LDL Receptor-Related Protein 6 (LRP6) [50,51,52]. PRMT1 and its enzymatic activity are involved in the activation/regulation of the canonical Wnt pathway in MDA-MB-468 TNBC cells [39] through the transcription of the two main components of the Wnt pathway, namely, LRP5 and Porcupine O-Acyltransferase (PORCN), by being recruited on their promoters (Figure 2).

Estrogen is associated with BC development as 80% of BCs express its receptor, Erα [53]. Erα signaling involves multiple actors in its nongenomic pathway [54,55]. In BC cells, PRMT1 is constitutively bound to Insulin-like Growth Factor 1 Receptor (IGF-1R). Insulin-like Growth Factor 1 (IGF-1) stimulation induces PRMT1 activation, thereby catalyzing Erα methylation leading to its interaction with IGF-1R and its phosphorylation, stabilizing their interaction. Then, IGF-1R phosphorylates IRS1 and Shc on tyrosine residues [56], forming docking sites for PI3K and GRB2, and activating Akt and ERK pathways, respectively [57] (Figure 2).

EMT is a cellular process in which epithelial cells lose their cell–cell adhesion and polarized organization and acquire a spindle-like morphology with enhanced cell migratory and invasive capacities, which encourages metastasis formation and is linked to therapeutic resistance [58,59,60]. By mediating the asymmetric dimethylation of histone H4R3 residue within the ZEB1 (zinc finger E homeobox-binding 1) promoter, PRMT1 activates ZEB1 expression and induces EMT in BC cells [23] (Figure 2). Asymmetric dimethylation of H4R3 by PRMT1 at the ZEB1 promoter is also described as promoting the acquisition of stem cell characteristics in BC cells and may contribute to the PRMT1-dependent inhibition of senescence in BC cells [23]. In addition, PRMT1 dimethylates the enhancer of *zeste* homolog 2 (EZH2) at R342 residue, which inhibits the phosphorylation of its T345 and T487 residues, attenuating EZH2 ubiquitylation [61], thereby decreasing the expression of EZH2 target genes and increasing EMT, cell invasion, and BC metastasis [62] (Figure 2).

In TNBC, PRMT1 sustains proliferative signaling by regulating the EGFR pathway [41]. In more than 70% of TNBC patients, EGFR overexpression is linked to poor clinical outcomes and metastasis [63]. PRMT1 regulates the EGFR signaling pathway by methylating EGFR on R198 and R200 residues. PRMT1-dependent EGFR methylation upregulates different signaling cascades, notably those involving Akt, ERK, or STAT3 in TNBC cells. [64]. In addition, PRMT1 regulates the EGFR signaling pathway at the transcriptomic level by being recruited to its promoter region and methylating histone H4, activating its transcription [39] (Figure 2). As EGFR-ERK signaling upregulates ZEB1 [65], PRMT1-mediated EGFR upregulation may also contribute to ZEB1 upregulation and ZEB1-mediated cancer stem cell regulation (Figure 2). 

### 3.3. Regulation of Alternative Splicing

Aberrant regulation of alternative splicing is tightly linked to BC occurrence and development and has become a target for cancer therapy [66,67,68]. For instance, the alternatively spliced BRCA1-Δ11q (partial skipping of exon 11) isoform is involved in breast tumorigenesis and the development of drug resistance [69]. HER2-Δ16 initiates a key oncogenic signal significantly affecting HER2-driven breast cancer stemness, tumorigenesis, and drug resistance [70]. Splicing analysis has revealed that PRMT1 expression is associated with alternative splicing disorders in breast tumor samples [71], and mass spectrometry-based proteomics on BC patient-derived xenograft (PDX) tumors has revealed that PRMT1 substrates include proteins involved in RNA processing [72]. PRMT1 methylome profiling identifies that PRMT1 methylates the splicing factor SRSF1 on R97 and R109 residues, which is critical for SRSF1 phosphorylation, SRSF1 binding with RNA, and exon inclusion. PRMT1-mediated SRSF1 methylation is important in regulating alternative splicing of genes that are known to be oncogenic, such as *AURKA* and *RAD1*, both highly expressed in breast tumor samples. In breast tumors, PRMT1 overexpression is associated with increased SRSF1 arginine methylation and aberrant exon inclusion, which are critical for breast cancer cell growth in vivo and in vitro [73]. High SRSF1 arginine methylation by PRMT1 promotes the exon inclusion of *AURKA* and *RAD1*, leading to the accumulation of the oncogene isoforms of these genes, which are involved in the regulation of cancer cell growth, metastasis, DNA repair, and drug resistance [74,75,76,77]. These results indicate that modulation of the RNA process by asymmetric dimethylation of proteins by PRMT1 is an indispensable biological process in BC cell lines and tumors. 

### 3.4. Steroid Receptors Regulation

Steroid receptors (SR) play many critical roles in human physiology and pathology. They are part of the nuclear hormone receptor family of transcription factors regulated by their ligand binding. Among them, estrogen receptor Erα and PR influence the development of the mammary gland. Dysregulation of their signaling and their overexpression leads to BC. Erα is a transcription factor that regulates many BC cell processes, including proliferation, differentiation, and survival. Besides their transcriptional effects, estrogen and Erα also regulate extra-nuclear events called non-genomic signaling [78]. In addition, numerous studies have established that PR is a key regulator of different aspects of breast tumorigenesis and tumor progression, such as cell migration and invasion [79,80]. In luminal BC, PRMT1 was reported to methylate Erα and PR [81,82]. Indeed, our group highlighted that Erα is methylated at R260 residue (mERα), in the DNA-binding domain of Erα by PRMT1 in response to estrogen or IGF-1 [57,82]. This event modulates the non-genomic estrogen signaling as it results in the assembly of a cytoplasmic complex containing mERα, the p85 subunit of PI3K, Src, and the focal adhesion kinase (FAK), promoting cellular proliferation and survival, by activating MAPK and Akt signaling cascades. This mERα/Src/PI3K complex is present in normal breast tissue and is hyperactivated in aggressive breast tumors, and it has been associated with a decrease in patient survival and with resistance to endocrine therapies (tamoxifen and fulvestrant) [83,84,85] (Figure 2).

In addition, our group demonstrated that PRMT1 regulates PR transcriptional activity both with and without hormone stimulation (Figure 2). Without progesterone, PRMT1 interacts with PR and some coregulators such as LSD1 and HP1γ in order to form a repressive complex, regulating the expression of a subtype of target genes involved in development and cell growth [86]. Moreover, in the absence of progesterone, PRMT1 allows the interaction between PR and the E3 ubiquitin ligase BRCA1. This interaction is essential for PR turnover and subsequently to finely tune the repression of genes regulated by progesterone before any stimulation [86].

Progesterone stimulation induces a dissociation of repressive complexes, the opening of chromatin, and the recruitment of coactivators including PRMT1. A transcriptomic study showed that PRMT1 inhibition decreases the expression of a subtype of PR target genes, controlling cell proliferation and migration [81]. The establishment of a stable cell line by Crispr-Cas9 technology where PR was mutated on R637 residue showed that PR methylation decreases the stability of the receptor, increasing its recycling and its transcriptional activity. Indeed, PRMT1 inhibition or loss of PR methylation at R637 decreases ERK activation and subsequently PR phosphorylation, a “primed” form of the receptor. Functionally, the absence of PR methylation decreases the oncogenic properties of PR, such as cell proliferation or colony formation, upon progesterone treatment. Likewise, low PRMT1 expression is predictive of longer patient survival in a subgroup of patients with high PR expression [81]. All these data indicate that PR methylation by PRMT1 acts as a molecular switch regulating transcription in mammary cells during tumorigenesis (Figure 2). 

### 3.5. Involvement in Double-Strand Break DNA Repair

Double-strand break (DSB) is a form of major DNA damage that challenges the integrity of the genome as it can lead, if left unrepaired, to mutagenic events, including loss of genetic material and chromosomal translocations. PRMT1 methylates several proteins involved in DNA damage repair regulation [87], and abrogation of its activity causes hypersensitivity to DNA damage, defects in cell cycle control, and an accumulation of chromosomal abnormalities. Of particular interest, PRMT1 targets MRE11 as well as 53BP1, both of which are critical for DNA repair pathway choice. Indeed, MRE11 initiates DNA end resection, thus promoting HR, whereas 53BP1 inhibits inappropriate resection of DNA ends during G1 to favor non-homologous end-joining (NHEJ). The methylation of MRE11 by PRMT1 on R587 residue is required for MRE11 to act as an exonuclease at sites of DNA DSBs and allows MRE11 to initiate a first step in HR [88]. Impaired methylation of MRE11 by PRMT1 has been shown to induce increased chromosomal instability, loss of DNA damage checkpoint activation, and a failure to recruit RPA and subsequently RAD51 to DSB ends [89]. 

PRMT1-mediated methylation of BRCA1 on R610 residue maintains DNA integrity during normal cell division. Depletion of PRMT1 promotes BRCA1 translocation from the nucleus to the cytoplasm and prevents BRCA1/BARD1 interaction and foci formation after irradiation in breast cancer cells, leading to an impaired DSB repair by HR [47]. By modulating the cellular localization of BRCA1, PRMT1 is an important regulator of the oncogenic functions of BRCA1, contributing to the epigenetic defense of breast cancer cells against ionizing radiation, and PRMT1-mediated methylation of BRCA1 is known to facilitate resistance to radiation therapy [47] (Figure 2). Furthermore, mass spectrometry analysis of PRMT1-interacting proteins identified USP11 as a PRMT1 substrate in BC cells [90]. USP11 is a deubiquitylating enzyme, a member of the ubiquitin-specific protease (USP) subfamily, previously shown to be implicated in HR modulation by positively regulating the formation and stabilization of the BRCA1–PALB2–BRCA2 complex to enable RAD51 loading onto ssDNA [91,92,93,94]. Methylation of USP11 by PRMT1 on R433 residue is important for RPA foci formation in late S/G2 cells and for RAD51 loading onto ssDNA at the G2/M checkpoint. Furthermore, deubiquitylation of PRMT1 by USP11 regulates methyltransferase activity and induces an increase in the methylation of MRE11 without affecting PRMT1 level. These results demonstrate that USP11 functions both as a substrate and a cofactor of PRMT1. Moreover, USP11 is implicated in breast cancer initiation and progression as it inhibits cell growth and survival by repressing Erα transcriptional activity. High USP11 expression is correlated with poor prognosis in Erα-positive patients [95]. 

Finally, PRMT1 regulates NHEJ by methylating 53BP1. 53BP1, a protein interacting with the tumor suppressor p53, translocates to DNA damage sites at an early stage of DNA repair and plays a crucial role in the choice of HR or NHEJ pathway [96,97]. Binding of 53BP1 at the DSBs protects DNA ends from resection to limit ssDNA formation, consequently promoting NHEJ [98]. Methylation of 53BP1 by PRMT1 regulates its recruitment to sites of DNA DSBs by enhancing its DNA binding activity and modulates its interaction with other DNA repair proteins [99,100]. PRMT1 was significantly correlated with therapeutic sensitivity to the PARP inhibitor Olaparib in BC cells [101]. Indeed, the authors demonstrated that PRMT1 confers resistance to Olaparib by modulating MYC signaling in TNBC. PRMT1 expression was associated with the MYC signature and was shown to regulate c-Myc protein stability by associating with and stabilizing c-Myc. Inhibition of c-Myc expression impaired HR gene expression and led to an increase in the sensitivity of TNBC cells to Olaparib [101,102].

### 3.6. Suppression of Tumor Immune Surveillance

Immune cells, which are present in the tumor microenvironment, play a key role in cancer metastasis [103]. Tumor immunosurveillance enables the immune system to eradicate tumor cells. Tumor cells escape this process by creating a tumor microenvironment that inhibits the body’s immune response and encourages tumor growth [104,105]. Regulatory T cells (Tregs), along with other tumor-infiltrating immune cells, contribute to this immunosuppressive microenvironment by inhibiting the activation of effector immune cells. Tumor cells acquire resistance to immune-mediated killing by downregulating antigen presentation, increasing immune checkpoint molecule expression, and inducing immune exhaustion. PRMT inhibitors can activate anti-tumor immunity and regulate immunotherapy efficacy.

PRMT1 expression was negatively correlated with the infiltration of CD8^+^ T cells and macrophages and was reversely correlated with the effector T cell signature in breast cancer [106]. PRMT1 acts as a suppressor of tumor immune surveillance in a cGAS (cyclic GMP-AMP synthase)-dependent manner through methylating cGAS at the conserved R133 residue on its N-terminus, and then blocking cGAS DNA sensing signaling, thus promoting tumor immune evasion [106]. Therapies that disrupt DNA damage repair to elevate cytosolic DNA levels synergize with anti-tumor immunotherapy by activating the cGAS pathway [107]. cGAS is a cytosolic DNA sensor that activates the type I interferon pathway and promotes tumor metastasis in breast cancer [108,109]. In addition, tumor-associated macrophages (TAMs) secrete the IL-6 cytokine, which triggers PRMT1 to mediate the formation of asymmetric dimethylation of the oncogene EZH2 at arginine 342, reinforcing EZH2 stability and leading to BC metastasis [62,110]. Moreover, high expression levels of IL-6 are positively correlated with PRMT1, meR342-EZH2, and EZH2 expression in BC patients. These results indicate that PRMT1 is a potential target for cancer immunotherapy, and PRMT1 inhibitor synergizes with immune checkpoint blockades to boost cancer immunity. 

### 3.7. Chemoresistance

PRMT1 has a divergent role in the regulation of BC cell apoptosis. In MCF7 cells, PRMT1 methylates the pro-apoptotic BAD (BCL2 antagonist of cell death) protein at R94 and R96 residues within the Akt consensus phosphorylation motif, promoting apoptosis [111]. In contrast, an inhibitory methylation mark by PRMT1 on the apoptotic signal-regulating kinase 1 (ASK1) at R78 and R80 residues prevents its activation and protects MCF7 and MDA-MB-231 cells from stress-induced apoptosis [112]. ASK1 plays a key role in cellular responses to various types of stress, including oxidative stress, by contributing to the activation stress-activated JNK and p38 signaling pathways, leading to the induction of apoptosis through caspase activation [113,114,115]. PRMT1, by methylating ASK1, suppresses paclitaxel-induced stimulation of ASK1 and apoptotic cell death in BC cells [112]. PRMT1-mediated methylation of ASK1 may therefore contribute to drug resistance in cancer cells. 

BRCA1 is frequently mutated in hereditary BC. In the absence of BRCA1 (or in cells with a dysfunctional BRCA1 protein), HR is suppressed, and the error-prone NHEJ takes over, which can lead to the accumulation of chromosomal aberrations and ultimately more tumorigenic cells [116]. Targeting poly (ADP-ribose) polymerase (PARP) has shown therapeutic potential in patients with TNBC who have a deficiency in the HR mechanism [117,118]. PRMT1 has been significantly correlated with therapeutic sensitivity to the PARP inhibitor Olaparib in BC cells [101]. PRMT1 confers resistance to Olaparib by modulating MYC signaling in TNBC. PRMT1 expression is associated with MYC signature and regulates c-Myc protein stability and c-Myc-mediated HR gene expression. Inhibition of PRMT1 downregulates c-Myc and sensitizes TNBC cells to Olaparib, suggesting that the methylation activity of PRMT1 may play a role in mediating resistance to PARP inhibitors [101,102]. 

Metabolic remodeling is a mechanism by which cancer cells acquire resistance. PRMT1 methylates 6-phosphofructo-2-kinase/fructose-2,6-bisphosphatase 3 (PFKB3), inducing the activation of phosphofructokinase 1 (PFK-1). PFKB3 methylation contributes to a metabolic shift towards the pentose phosphate pathway (PPP) to accelerate the synthesis of NADPH, a reducing equivalent for recycling reduced glutathione (GSH), which helps chemoresistance [119]. PRMT1 also upregulates the serine synthesis pathway (SSP) in chemo-resistant TNBC cells [120]. In these cell lines, PRMT1 increases arginine methylation of PFKB3, PKM2 (pyruvate kinase M2), and PHGDH (3-phosphoglycerate dehydrogenase. These three enzymes (PHGDH, PFKB3, and PKM2) catalyze the biotransformation of glucose into SSP, which branches out of glycolysis to deliver a substrate for the survival of TNBC cell lines. In addition, PRMT1 simultaneously activates de novo fatty acid synthesis through the production of α-ketoglutarate (α-KG) via activation of PHGDH by arginine methylation at R20 and R54. A-KG is a source of fatty acid synthesis to produce citrate and acetyl-CoA in breast cancer cells [121]. 

## 4. PRMT5 in Breast Cancer

PRMT5 is the major type II protein arginine methyltransferase and regulates a wide range of histones and non-histone substrates [122,123]. The structure of PRMT5 includes an N-terminal TIM barrel domain, a middle Rossman fold domain, and a C-terminal β-barrel domain. PRMT5 commonly associates with the methylosome protein 50 (MEP50) in an octameric complex of four PRMT5 and four MEP50. The PRMT5-MEP50 complex stabilizes and potentiates PRMT5 methyltransferase activity [124]. PRMT5 is responsible for the methylation of various proteins like histones H2A, H3, and H4, transcription factors, and cell receptors [125,126]. PRMT5 regulates a broad range of biological processes, including differentiation, chromatin regulation, splicing, DNA damage response, cellular growth and development, and cell signaling [127]. Several studies have highlighted the role of PRMT5 as a tumor-promoting factor in several types of cancers, including colon, breast, lung, leukemia, and pancreatic cancers among others [128,129,130,131,132]. PRMT5 is upregulated in BC tissues compared to normal tissues, and its high expression is associated with decreased RFS for all subtypes except HE2+ tumors [133,134] (Figure 3). 

PRMT5 regulates key BC pathways, including cell cycle, EMT, DNA damage repair, cancer stem cell maintenance [24,129,135,136], and gene expression through histone modification and chromatin remodeling [129,136,137,138] (Table 2). 

### 4.1. Regulations of Oncogenic Pathways

Krüppel-like factor 4 (KLF4, GKLF) is a major regulator of DNA damage response, inflammation, apoptosis, and stem cell renewal [139,140]. KLF4 has been reported to act as an oncogenic factor in breast tumors [141,142]. The Von Hippel–Lindau tumor suppressor (pVHL), a ubiquitin protein ligase, regulates the proteasome-dependent degradation of KFL4. PRMT5 physically interacts with KLF4 to catalyze its methylation on R374, R376, and R377 residues, altering KLF4 conformation and blocking its ubiquitylation by pVHL, thereby stabilizing KLF4 [143]. Stabilization of KLF4 in BC cells increases oncogenic pathways such as EGF/EGFR, MAPK, and CDK1, promoting cell growth in mammary gland and breast tumor cells [144] (Figure 4). PRMT5 has also been shown to interact with Krüppel-like factor 5 (KLF5), an oncogenic factor highly expressed in basal-like BC [145,146]. PRMT5 catalyzes the dimethylation of KLF5 at R57 residue, inhibiting its phosphorylation, ubiquitylation, and degradation by FBW7, promoting BC cell proliferation [147] (Figure 4). 

The liver X receptor α (LXRα) acts as a tumor suppressor gene in a variety of tumors and in BC. Downregulation of LXRα is often observed in BC patients, and its loss increases proliferation, invasion, and aerobic glycolysis. LXRα also regulates the expression of the NF-κBp65 subunit, a major actor in aerobic glycolysis of BC cells [148]. PRMT5 has been shown to regulate the expression of LRXα in BC cells and mouse models and promotes BC cell proliferation, invasion, and aerobic glycolysis in an NF-κBp65-dependent manner [149] (Figure 4). Wnt/β-catenin signaling is known to be activated in almost 50% of BC patients [150]. In TNBC cells, PRMT5 promotes Wnt/β-catenin proliferative signaling through transcriptional repression of DKK1 and DKK3. PRMT5 interacts with the promoter regions of DKK1 and DKK3 and catalyzes H3R8 and H4R3 symmetric methylation. PRMT5 is recruited to the promoter regions of the two Wnt/β-catenin antagonists DKK1 and DKK3 and regulates H3R8 and H4R3 symmetric methylation in their promoter regions. PRMT5-silencing of DKK1 and DKK3 leads to enhanced expression of c-MYC, CYCLIN D1, and SURVIVIN, and BC cell proliferation [137] (Figure 4).

### 4.2. Maintenance of Breast Cancer Stem Cells

Breast cancer stem cells (BCSCs) are a small subpopulation of heterogeneous BC cells with self-renewal and proliferation properties. BCSCs are believed to be the driving force behind BC tumor initiation, progression, metastasis, and drug resistance [151]. Previously, PRMT5 has been shown to contribute to stem cell function in leukemia and glioblastoma [152,153]. In BC cells, PRMT5 is recruited to the Forkhead Box P1 (FOXP1) promoter, a winged helix/forkhead transcription factor associated with cancer stem cell function [154] and facilitates H3R2 methylation. H3R2 methylation triggers the recruitment of the SET domain-containing 1 (SET1) histone methyltransferase via WDR5, leading to H3K4 trimethylation and gene expression. Increased levels of PRMT5 and FOXP1 induce BCSC proliferation and self-renewal (Figure 4). Moreover, depletion of PRMT5 in established tumors decreases BCSC frequency, highlighting its role in BCSC maintenance [129]. Furthermore, stabilization of KLF4 by PRMT5 in BC cells increases the expression of several crucial stem cell factors such as Myc, Sox9, and BMI1, suggesting a role of PRMT5/KLF4 in the maintenance of BCSCs [144].

### 4.3. Regulation of Epithelial–Mesenchymal Transition

PRMT5 has been shown to regulate EMT in tumors [155,156,157]. In BC cells PRMT5 interacts with Snail Family Transcriptional Repressor 2 (SNAI2) and lysine demethylase 1A (KDM1A) to modulate the expression of E-cadherin and vimentin, two major EMT molecular markers. SNAI2 interacts with PRMT5 and KDM1A, and the tri-complex is recruited to the E-cadherin promoter region. PRMT5 catalyzes H4R3me2s, and KDM1A mediates the demethylation of H3K4me2 on the E-cadherin promoter, inhibiting E-cadherin expression. Furthermore, the tri-complex is also recruited to the vimentin promoter region to function as a transcriptional coactivator. In this case, PRMT5 catalyzes H3R2me2s, and KDM1A removes H3K9me2 on the vimentin promoter (Figure 4). These results show that PRMT5 and KDM1A act as a dual epigenetic regulator on the promoters of E-cadherin and vimentin and cooperate to promote the EMT and invasion of BC cells. TGFβ signaling is one of the major pathways regulating the EMT process and is frequently associated with tumor progression and metastasis dissemination. TGFβ simulation of BC cells promotes increased PRMT5, an increase in H3R2me1, H3R2me2s, and H4R3me2s histone methylation, and an altered expression of EMT markers such as E-cadherin, vimentin, and SNAI2 [158].

### 4.4. Splicing Regulation

The role of PRMT5 in splicing regulation has been the focus of several studies. PRMT5 regulates the assembly of the spliceosome, responsible for removing introns from pre-mRNA transcripts in the nucleus [159]. PRMT5 catalyzes symmetric dimethylation motifs in Sm B/D1/D3, promoting the loading of Sm proteins onto the SMN complex, a necessary step the spliceosome assembly [160,161]. In addition, PRMT5 substrates and interactors include spliceosomal proteins and pre-mRNA splicing factors, and inhibiting PRMT5 leads to global alteration in AS events [162,163,164]. PRMT5 depletion or inhibition causes AS of MDM4, leading to the activation of the p53 pathway and apoptosis in different models [165,166,167]. PRMT5 also regulates the splicing of activating transcription factor 4 (ATF4), and its loss decreases ATF4 expression and increases oxidative stress [168]. The methylation of the SRSF1 splicing factor by PRMT5 affects SRSF1 binding to splice sites, resulting in broad changes in splicing programs [164]. In BC cells, MEP50 interacts with the splicing factor ZNF326, and PRMT5 catalyzes its symmetric demethylation at R175 [169]. Loss of PRMT5 induces alteration in AS events such as the inclusion of A-T rich exons, an AS defect resulting from the loss of ZNF326 [169]. Through its roles in the spliceosome assembly and the regulation of splicing factor functions, PRMT5 controls splicing events in cells, and its loss or inhibition causes major splicing defects.

### 4.5. Regulation of Immunotherapy

Although the immune system can remove aberrant cells [170], cancer cells use a variety of mechanisms to escape immune surveillance to proliferate, including recruitment of immunosuppressive cells, such as regulatory T cells (Tregs) or TMA [171,172]. PRMT5 has been shown to decrease the level of type I interferon (IFN) and chemokine production and repress antigen presentation, suppressing cancer cell recognition by T cells. PRMT5 can attenuate the cGAS-STING pathway by dimethylating interferon-gamma inducible protein 16 (IFI16), a cGAS complex component [173,174]. Using a mass spectrometry approach, the forkhead family transcription factor FOXP3 was identified as a PRMT5 substrate. Methylation of FOXP3 by PRMT5 is necessary for its activation to maintain Treg function [175]. Treg cells infiltrate into tumor tissues and impair anti-tumor immune responses [176]. In human tumors, tumor infiltration with FOXP3+ Tregs is associated with poor prognosis [177].

In TNBC, PRMT5 catalyzes the methylation of Kelch-like ECH-associated protein 1 (KEAP1) [178]. KEAP1 methylation by PRMT5 increases its stability and inhibits the downstream NRF2/HMOX1 pathway, critical in the regulation of ferroptosis [179]. In addition, high expression of PRMT5 indicated strong resistance in TNBC to immunotherapy, and PRMT5 inhibitors potentiated the therapeutic efficacy of anti-PD-1 immunotherapy in mice [178]. By regulating KEAP1, NRF2, and HMOX1 expression levels, PRMT5 promotes resistance to immunotherapy-induced ferroptosis in breast cancer.

### 4.6. Involvement in Double-Strand Break DNA Repair

PRMT5 plays a fundamental role in DNA damage repair. Histone arginine methylation by PRMT5 induces the expression of DDR genes. PRMT5 increases the expression of *DDR* genes involved in HR and NHEJ pathways, such as RAD51 and BRCA1/2 [180]. PRMT5 also regulates the expression of DDR proteins by regulating RNA splicing [181]. In BC, after doxorubicin treatment, PRMT5 promotes BRCA1 expression by attenuating m6A methylation of BRCA1 mRNA and enhancing its stability [182]. PRMT5 substrates include multiple DDR proteins, and their functions, stability, DNA-binding ability, and interaction are regulated by PRMT5-catalyzed symmetrical demethylation of arginine residues. PRMT5 regulates the methylation of p53, a central protein in the DDR that maintains genome stability, on R335 and R337 [12]. Methylation of p53 enhances its function in DNA damage response and promotes the expression of other DDR genes [183]. 

In BC, PRMT5 regulates the methylation of KLF4 at three arginine residues (R374, R376, and R378) [184]. Interestingly, mutation of these three residues increases foci formation and the presence of 53BP1 at DNA damage sites, suggesting a defect in HR and enhanced NHEJ [144]. Knockdown of PRMT5 or KLF4 mutant impairs BRCA1 foci formation and suppresses HR pathways [185]. PRMT5-mediated methylation of KLF4 facilitates DSB repair by HR. 

Preclinical studies suggest PRMT5 regulation of DDR pathways may be a therapeutic opportunity, and PRMT5 inhibition may increase the sensitivity of cancer cells to PARP inhibition [180,186,187]. PRMT5 inhibition increases the expression of DNA damage protein in breast cancer cell lines. A combination of PRMT5 inhibitor (GSK3326595) and PARP inhibitor, (niraparib) produced a synergistic effect on growth inhibition of BC cell lines and patient-derived xenograft (PDX) models. In mice, a combination of the two treatments resulted in a diminution of tumor growth [188]. In addition, inhibition of PRMT5 using a potent and highly selective small-molecule inhibitor (C220) correlates with downregulated DDR genes such as BRCA1, RAD51, and ATM in BC cell lines. PRMT5 inhibition reduces the presence of PRMT5 and H4R3me2s on promoter regions of DDR genes and regulates DDR gene expression by promoting exon skipping and intron retention. The combination of PRMT5 inhibitors and olaparib inhibits PDX growth both in vitro and in vivo. Furthermore, PRMT5 inhibition was shown to significantly inhibit the growth of olaparib-resistant tumors [136]. These studies reveal the beneficial combinatorial effects of PRMT5 inhibition with other therapies.

### 4.7. Dual Role of PRMT5

Growing evidence shows that different subcellular localizations of PRMT5 within cells determine its effect: cytoplasmic expression of PRMT5 seems to be associated with less-differentiated cells, while nuclear PRMT5 correlates with cell differentiation and cell cycle arrest. In self-renewing embryonic stem cells, cytoplasmic PRMT5 plays an essential role in maintaining an undifferentiated state, while it relocates to the nucleus upon differentiation [189,190].

Although PRMT5 is a major cytoplasmic protein, it appears to also be expressed in the nucleus. This dual localization suggests that in each compartment the PRMT5 interactome is different, thus regulating distinct molecular programs associated with diverse outcomes. For instance, in prostate epithelial cells, PRMT5 was shown to be predominantly nuclear, whereas it was mainly in the cytoplasm of prostate cancer cells [191]. This observation was confirmed in other tissues such as lung and skin. In BC, the cytoplasmic expression of PRMT5 is higher in the aggressive form of BC, such as TNBC, whereas it is mostly nuclear in luminal BC [192]. Likewise, our team showed that a high expression of PRMT5 in the nucleus is associated with better patient survival [193]. More recently, using two retrospective cohorts of unselected BC, our team showed that the nuclear expression of PRMT5 is a predictive marker of sensitivity to tamoxifen but was not correlated to the response to aromatase inhibitors [194]. In addition, in tamoxifen-sensitive tumors, tamoxifen triggers PRMT5 nuclear translocation, where it methylates ERα in the DNA-binding domain, a prerequisite for the recruitment of transcriptional corepressors such as SMRT and HDAC1 which participate in the inhibition of the transcription of ERα target genes [25] (Figure 5). 

Conversely, in TNBC glucocorticoids trigger the interaction of the glucocorticoid receptor (GR) with PRMT5 on chromatin, a key event for the recruitment of HP1γ that facilitates the binding of the activated form of RNA polymerase II. This complex regulates the expression of GC-dependent target genes involved in cell migration in vitro and in vivo. Even though PRMT5 has been demonstrated to methylate GR [193], in this context PRMT5 regulates cell migration independently of its catalytic activity, acting instead as a scaffold protein [194] (Figure 5).

From a mechanistic point of view, little information is available concerning the regulation of the nucleocytoplasmic translocation of PRMT5, particularly in BC cells. PRMT5 is most frequently localized in the cytoplasm of the cells, likely due to the fact that it possesses no nuclear localization signal (NLS) but three nuclear export signals (NES) [191]. In osteosarcoma cells, PRMT5 is sent into the nucleus by the EMT transcription factor SNAIL associated with the transcriptional corepressor AJUBA to repress the transcription of E-cadherin [195]. In lung cancer cells, Akt activation promotes the transport of PRMT5 from the nucleus to the cytoplasm, although the mechanisms underlying this process have not been elucidated [196]. 

## 5. Targeting PRMTs in Breast Cancer

PRMT inhibitors were initially reported in the early 2000s, and recently specific inhibitors were designed to specifically target PRMT3, PRMT4, PRMT5, and PRMT6 in preclinical cell culture animal models, revealing the therapeutic and pharmacological potential of targeting PRMTs in cancer [197,198]. Significant efforts have been devoted to identifying, synthesizing, and applying PRMT inhibitors as potential cancer treatments. PRMT inhibitors, particularly those targeting PRMT1 and PRMT5 due to their well-documented role in cancer progression, are being investigated as therapies for hematological malignancies and solid tumors and have progressed to clinical trials [199,200,201] (Table 3). 

### 5.1. PRMT1 Inhibitors

For a long time, the inhibitors available for PRMT1 lacked specificity, efficacy, and bioavailability. GSK3368715 is a powerful and reversible inhibitor that targets type I PRMTs by binding to the protein substrate pocket in a SAM-uncompetitive manner. This inhibition decreases intracellular levels of ADMA and results in the accumulation of MMA and SDMA. In preclinical cancer models, including BC, GSK3368715 significantly reduced global ADMA levels and demonstrated strong anti-proliferative effects across various tumor types. While targeting PRMT1 seems to be a promising therapeutic strategy, the phase 1 clinical trial for GSK3368715 was terminated early due to a lack of clinical efficacy, extensive treatment-related side effects, and dose-limiting toxicities [199]. 

ZJG51 and ZJG58 were synthesized using molecular modeling to target and occupy a sub-binding pocket of PRMT1. These two compounds display significant inhibitory effects on PRMT1 and inhibitory properties on the growth of tested cancer cell lines, with ZJG51 having relatively stronger activities against cancer cells. ZJG51 inhibited migration of HeLa cells and induced apoptosis by regulating PRMT1-related proteins. ZJG51 was shown to stably bind to PRMT1. Its mechanism of action may involve the activation of caspase 9 and inhibition of EMT [202].

TC-E 5003, also known as N,N′-(Sulfonyldi-4,1-phenylene)bis(2-chloroacetamide), is a selective inhibitor of PRMT1. It has successfully been used to modulate the inflammatory response by reducing lipopolysaccharide (LPS)-induced nitric oxide (NO) production and inflammatory gene expression and by downregulating key signaling pathways such as NF-κB and AP-1 [203]. Additionally, TC-E 5003 exhibits strong anti-tumor effects in vitro in the context of lung and breast cancer cell lines. Using an injectable, in situ-forming implant system (INEI) to deliver TC-E 5003 was reported to significantly enhance its anti-tumor efficacy in animal models, reaching 68.23% growth rate inhibition of xenografted human lung cancer cells compared to 31.76% with TC-E 5003 alone [204]. These findings suggest that TC-E 5003 may be a highly promising molecule not only with anti-inflammatory affects but also as a broad-spectrum anti-tumor drug, with improved druggability when delivered via the INEI system.

### 5.2. PRMT5 Inhibitors

Various PRMT5 inhibitors have been developed, demonstrating potent anticancer effects in both preclinical and clinical settings. Key inhibitors such as EPZ015938 (GSK3326595) and JNJ-64619178 have been reported to attenuate cancer growth in cell and animal models, to display tumor inhibition and regression in biomarker-driven xenograft models, and to maintain tumor growth inhibition after treatment. These inhibitors are currently undergoing clinical trials for various cancers, including solid tumors, non-Hodgkin’s lymphoma (NCT02783300), myelodysplastic syndrome (MDS), and acute myeloid leukemia (AML) (NCT03614728). 

CMP5, the first PRMT5-specific inhibitor, was identified by screening the ChemBridge CNS-Set library of small molecules and shows reduced tumor cell viability by inhibiting PRMT5 [205]. In BC cells, PRMT5 inhibition by CMP5 reduces its recruitment and histone methylation at specific promoter regions, leading to decreased expression of cyclin D1 and survivin. This inhibition alters the growth characteristics of BC cells and induces their death [204,206]. Additionally, co-inhibition of cyclin-dependent kinase 4/6 (CDK4/6) and PRMT5 was reported as an effective and well-tolerated treatment strategy [207]. 

Another PRMT5 inhibitor, EPZ015666, identified through a homogeneous time-resolved fluorescence assay, disrupts the MEP50 complex crucial for PRMT5 methyltransferase activity. EPZ015666 is the first orally bioavailable and highly selective PRMT5 inhibitor, showing anti-proliferative effects in both in vitro and in vivo models [208]. PRMT5 degrader MS4322, developed using proteolysis-targeting chimera (PROTAC) technology, links EPZ015666 to a von Hippel–Lindau (VHL) E3 ligase ligand, reducing PRMT5 expression, and has shown promising results in a mouse pharmacokinetic study [209]. Combining PRMT5 inhibitors with existing chemotherapeutics has proven beneficial in different cancer types. For instance, the combination of EPZ015666 with gemcitabine enhances the DNA-damaging effect of gemcitabine in pancreatic ductal adenocarcinoma (PDAC) cells, helping to overcome therapeutic resistance [210]. In BC cells, combining EPZ015666 with chemotherapeutic agents like etoposide or cisplatin leads to a synergistic effect, improving cell viability [211]. 

More recently, the PRMT5-MTA complex emerged as a promising drug target for treating MTAP-deleted cancers. MRTX1719, a potent and selective inhibitor of the PRMT5-MTA complex, was discovered to selectively inhibit PRMT5 activity in MTAP-deleted cells. In tumor xenograft-bearing mice, daily oral administration of MRTX1719 led to dose-dependent inhibition of PRMT5-dependent protein modification, correlated with anti-tumor activity [212].

## 6. Conclusions

BC remains a prevalent and challenging disease, characterized by its molecular heterogeneity dictating distinct treatment approaches for its various subtypes. While therapies targeting the estrogen receptor, progesterone receptor, and HER2 have significantly improved patient outcome, resistance remains a major challenge. 

This review highlights the critical roles of PRMTs, specifically PRMT1 and PRMT5, in BC initiation, progression, dissemination, and treatment resistance. PRMT1 is implicated in numerous oncogenic pathways, including transcriptional regulation, DNA repair, and cell signaling, promoting breast cancer cell proliferation, invasion, and resistance to therapy. Similarly, PRMT5 influences gene expression, cell cycle control, and maintenance of cancer stem cells, contributing to tumor progression and poor prognosis. The dual role and localization of PRMT5 in BC further underscores its complex involvement in tumor biology. Our group has shown that nuclear PRMT5 is associated with differentiated cells and better prognosis, while cytoplasmic PRMT5 is linked to less differentiated, aggressive cancer cells [194]. This localization-dependent function of PRMT5 presents potential avenues for targeted therapies. Emerging results show that specific inhibitors targeting PRMTs display promising results in a large range of preclinical cancer models, reducing tumor growth and sensitizing BC cells to therapies. Targeting PRMTs could provide new therapeutic strategies to overcome resistance to treatment and improve clinical outcome for breast cancer patients. Future research should focus on developing and refining PRMT inhibitors and further characterize the mechanisms regulating PRMT activity and localization. Moreover, the combination of PRMT inhibitors with current therapeutics could enhance their efficacy. Overall, targeting PRMT1 and PRMT5 represents a promising therapeutic strategy in BC to improve patient clinical outcome. 

## Figures and Tables

**Figure 1 ijms-25-08854-f001:**
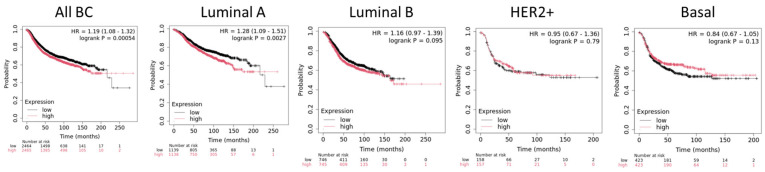
**Association of PRMT1 expression and relapse-free survival (RFS) in breast cancer subtypes.** Kaplan–Meier curves created by the public database and web application *KM plotter.* Molecular subtypes are based on St Gallen criteria [33].

**Figure 2 ijms-25-08854-f002:**
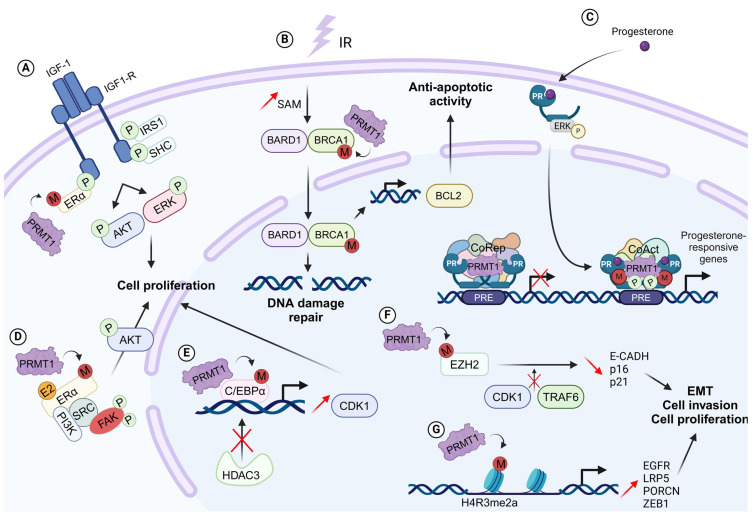
**PRMT1 in breast cancer.** (**A**). Upon IGF-1 stimulation, PRMT1 methylates Erα. IGF-1R phosphorylates IRS1 and SHC on tyrosine residues leading to the recruitment of PI3K and GRB2, activating AKT and ERK pathways. (**B**). PRMT1 controls the cellular localization of BRCA1 and facilitates DNA homologous recombination and upregulates the anti-apoptotic protein BCL2. (**C**). Without progesterone, PRMT1 interacts with PR and forms a repressor complex with other corepressors to regulate the expression of a subtype of target genes. After progesterone stimulation, chromatin opening allows the recruitment of PR coactivators, including PRMT1 which methylates PR on arginine 637 to regulate the expression of progesterone responsive genes. (**D**). PRMT1 methylates ERα at R260 residue in response to estrogen, resulting in the formation of a complex containing mERα, the p85 subunit of PI3K, SRC, and the focal adhesion kinase (FAK), and activating MAPK and AKT signaling cascades. (**E**). PRMT1-dependent methylation of C/EBPα promotes expression of cyclin D1 by blocking the interaction between C/EBPα and its corepressor HDAC3. (**F**). Methylation of EZH2 by PRMT1 regulates its stability and promotes EMT and breast cancer metastasis. (**G**). PRMT1 is recruited to the promoter region of EGFR, LRP5, PORCN, and ZEB1 and catalyzes H4R3 methylation to promote genes expression.

**Figure 3 ijms-25-08854-f003:**
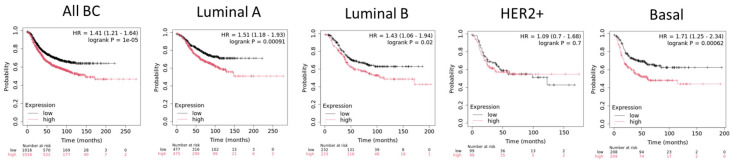
Association of PRMT5 expression and relapse-free survival (RFS) in breast cancer subtypes. Kaplan–Meier curves created by the public database and web application KM plotter. Molecular substypes are based on St Gallen criteria [33].

**Figure 4 ijms-25-08854-f004:**
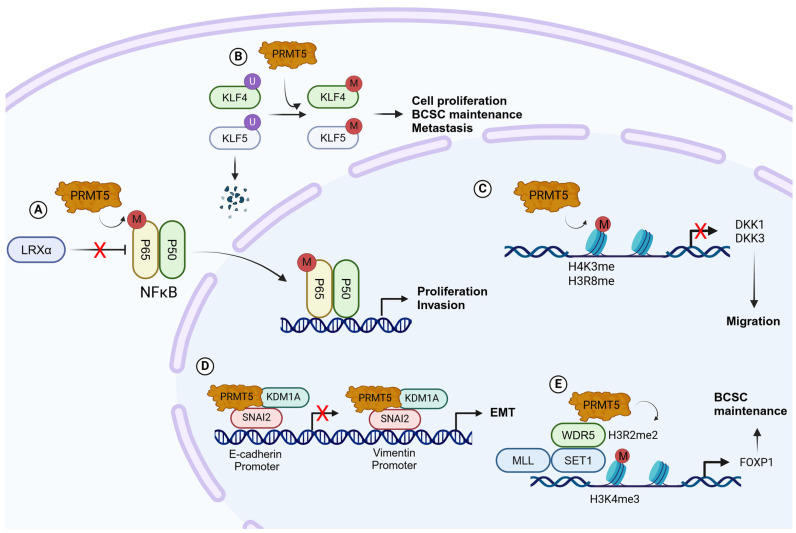
**PRMT5 in breast cancer** (**A**). PRMT5 promotes aerobic glycolysis and invasion of BC cells by regulating the LXRα/NF-κBp65 pathway. (**B**). PRMT5 interacts with KLF4 and KLF5 and regulates their methylation and stability. Stabilization of KLF4 and KLF5 in BC cells increases the expression of oncogenic pathways such as EGF/EGFR, MAPK, and CDK1 and cell proliferation, BCSC maintenance, and metastasis. (**C**). PRMT5 activates Wnt/β-catenin signaling via epigenetic silencing of DKK1 and DKK3. (**D**). SNAI2 interacts with PRMT5 and KDM1A and is recruited to the E-cadherin promoter region. PRMT5 catalyzes H4R3me2s, and KDM1A mediates the demethylation of H3K4me2, inhibiting E-cadherin expression. PRMT5 catalyzes H3R2me2s, and KDM1A removes H3K9me2 on the vimentin promoter to function as a transcriptional coactivator. (**E**). PRMT5 recruitment to the FOXP1 promoter facilitates H3R2me2s, SET1 recruitment, H3K4me3, and FOXP1 gene expression.

**Figure 5 ijms-25-08854-f005:**
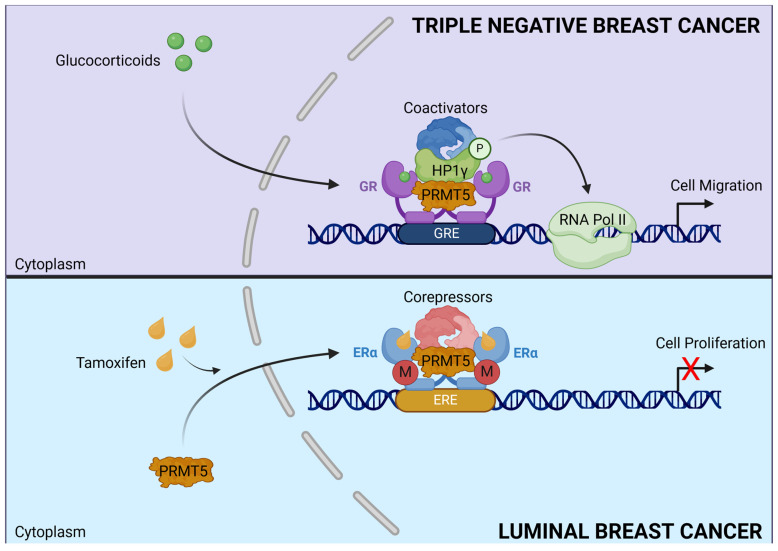
**Dual role of PRMT5 in breast cancer.** In TNBC, upon GC treatment, PRMT5 triggers the interaction between GR and HP1γ regulating cell migration independently of its enzymatic activity. In luminal BC, tamoxifen induces PRMT5 translocation and ERα methylation to recruit corepressor complexes that inhibit transcription.

**Table 1 ijms-25-08854-t001:** PRMT1 substrates in breast cancer.

Substrate	Residue	Functional Impact	Refs.
H4	R3	Transcriptional regulation	[41,42]
P53	N/A	Inhibits p53 transcriptional activity	[43]
C/EBPα	R35, R156, R165	Promotes C/EBPα transcriptional activity	[44]
EGFR	R198, R200	Enhances receptor signaling	[40,42]
EZH2	R342	Inhibits EZH2 transcriptional activity	[45]
Erα	R260	Promotes MAPK and AKT signaling	[46]
PR	R637	Regulates stability and transcriptional activity	[47,48]
MRE11	R587	Promotes DNA end resection	[37]
BRCA1	R610	Regulates BRCA1 subcellular localization	[49]
USP11	R433	RPA foci formation and RAD51 regulation	[50]
53BP1	ND	Enhances DNA repair process	[51]
BAD	R94, R96	Inhibits BAD anti-apoptotic activity	[52]
ASK1	R78, R80	Anti-apoptotic function and drug resistance	[31]

**Table 2 ijms-25-08854-t002:** PRMT5 substrates in breast cancer.

Substrate	Residue	Functional Impact	Refs.
H3	R8, R2	Transcriptional regulation	[94,103,104,105]
H4	R3	Transcriptional regulation	[103,105]
NF-κB (p65)	R30	Enhances p65 binding to DNA and transcriptional activity	[106]
KLF4	R376, R377	Protein stabilization	[107]
KLF5	R57	Protein stabilization	[108]
GR	ND	Promotes GR transcriptional activity	[109]
ERα	ND	Inhibits ERα transcriptional activity	[23]

**Table 3 ijms-25-08854-t003:** PRMT inhibitors.

Target	Drug	Mechanism of Action	Phase/Status	Intended Use
Type I PRMTs	GSK3368715	Substrate competitive	Phase I; terminated	Single agent
PRMT1	ZJG51	Substrate competitive	Preclinical models	Single agent
	TC-E 5003	Substrate competitive	Preclinical models	Single agent
PRMT5	GSK3326595	Substrate competitive	Phase II; terminated	Single agent
			Phase I; active	Combination with pembrolizumab
			Preclinical models	Combination with palbociclib
	JNJ-64619178	Dual SAM/substrate competitive	Phase I; active	Single agent
	CMP5	SAM-competitive	Preclinical models	Single agent
	EPZ015666	Substrate competitive	Preclinical models	Single agent
			Preclinical models	Combination with erlotinib
			Preclinical models	Combination with paclitaxel
	MRTX1719	PRMT5-MTA inhibitor	Phase I/II; recruiting	Single agent

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
