# Peer review of "Role of PRMT1 and PRMT5 in Breast Cancer"

_ijms, 2024, doi:10.3390/ijms25168854_

Round 1

Reviewer 1 Report

Comments and Suggestions for Authors

Reviewer 2 Report

Comments and Suggestions for Authors

The authors conduct a comprehensive review of current literature delineating the role of Protein Arginine Methyltransferases 1 and 5 (PRMT 1&5) in breast cancer. They summarize the findings from research conducted by their group and others, categorize the effects and targets of PRMT1 and 5-mediated methylation in breast cancer, and review the current state of PRMT1/5 targeted therapy.

The review flows well, includes informative graphical summaries and discusses recent developments.

Major comments:

1.        Since the review specifically discusses breast cancer, it would be useful to include Kaplan Meier curves for association of PRMT1 and 5 with prognosis in different subtypes of breast cancer (PFS, OS).

2.        The manuscript would be strengthened by including a summary table outlining the PRMT1/5 inhibitors discussed in sections 5.1 and 5.2 categorized by MOA, stages of development i.e preclinical/clinical trials, intended use as monotherapy or combination with chemo/radiation/surgery/immunotherapy etc.

3.        The manuscript would be further strengthened by also summarizing emerging literature on other roles of PRMT1/5 in breast cancer including alternative splicing, effect on immunotherapy efficacy, chemoresistance etc.

Minor comments:

1.        Correct the spelling to “inhibits” in Table 1

Comments on the Quality of English Language

Minor comment:

1.        Correct the spelling to “inhibits” in Table 1
